# An Evaluation of Sun-Glint Correction Methods for UAV-Derived Secchi Depth Estimations in Inland Water Bodies

Edvinas Tiškus * 🔟, Martynas Bučas 🔟, Diana Vaičiūtė, Jonas Gintauskas 🔟 and Irma Babrauskienė

Marine Research Institute, Klaipeda University, 92294 Klaipeda, Lithuania; martynas.bucas@ku.lt (M.B.);
diana.vaiciute@ku.lt (D.V.); jonas.gintauskas@ku.lt (J.G.); irma.babr@gmail.com (I.B.)
* Correspondence: edvinas.tiskus@ku.lt

**Abstract:** This study investigates the application of unoccupied aerial vehicles (UAVs) equipped with a Micasense RedEdge-MX multispectral camera for the estimation of Secchi depth (SD) in inland water bodies. The research analyzed and compared five sun-glint correction methodologies—Hedley, Goodman, Lyzenga, Joyce, and threshold-removed glint—to model the SD values derived from UAV multispectral imagery, highlighting the role of reflectance accuracy and algorithmic precision in SD modeling. While Goodman's method showed a higher correlation (0.92) with in situ SD measurements, Hedley's method exhibited the smallest average deviation (0.65 m), suggesting its potential in water resource management, environmental monitoring, and ecological modeling. The study also underscored the quasi-analytical algorithm (QAA) potential in estimating SD due to its flexibility to process data from various sensors without requiring in situ measurements, offering scalability for large-scale water quality surveys. The accuracy of SD measures calculated using QAA was related to variability in water constituents of colored dissolved organic matter and the solar zenith angle. A practical workflow for SD acquisition using UAVs and multispectral data is proposed for monitoring inland water bodies.

**Keywords:** UAVs; Secchi depth; multispectral imagery; sun glint; quasi-analytical algorithm; remote sensing

## 1. Introduction

Secchi depth (SD), an essential measure of water transparency in aquatic ecosystems, provides a critical indication of water quality and ecological health [1–3]. In Europe, SD is important for following the water quality rules set by the European Water Framework Directive (WFD), which mandates member states to uphold ecological standards [4]. SD thus helps identify impairments affecting water quality and aids in devising and executing management plans to protect water bodies [5].

The Secchi disk is the main tool used for obtaining SD measurements, particularly in vast and complex aquatic ecosystems [6–8]. However, traditional methods—visual observations using a white Secchi disk—can be labor-intensive, time-consuming, and potentially influenced by observer bias [9,10]. They also typically offer point-level measurements, which may not reflect water transparency over larger areas [11,12].

The limitations of traditional methods have spurred interest in remote sensing techniques for estimating SD in complex aquatic ecosystems. Techniques ranging from satellite data to unoccupied aerial vehicles (UAVs) equipped with light detection and ranging (LiDAR) systems and hyperspectral imagery have been explored for accuracy [13]. These techniques promise cost-effective and efficient high-resolution spatial data on water transparency over large areas, with multispectral imagery from UAVs emerging as a promising alternative [14].

UAVs can provide higher-resolution images than satellites, allowing for more precise measurements of water bodies, including small and shallow ponds [15,16], which are

often not well captured by satellite imagery [16,17]. Furthermore, UAVs can be flown at a specific time and location, which is particularly important for SD measurements, as the measurement of water transparency can be influenced by various environmental factors, such as partial cloud cover, water turbidity, and the angle of the sun [18,19]. Although weather conditions influence the ability of UAV flight (with Inspire 2 UAV if winds exceed around 10 m/s), UAVs can collect data during fully cloudy conditions, where optical satellite data are heavily affected by clouds. Additionally, they can be equipped with diverse sensors for capturing multispectral or hyperspectral images, thus providing valuable data on water quality and ecology [14,20]. Furthermore, UAVs can serve as ground truth to validate satellite data [21,22], thereby enhancing accuracy for the monitored area, e.g., one tile coverage of Sentinel-2 can reach from 100,000 ha, while UAVs cannot cover relatively large areas. Moreover, UAV technology presents a strategic advantage in regions where conventional methodologies struggle due to logistical complications.

SD is primarily influenced by three optical components: Chl-a, colored dissolved organic matter (CDOM), and total suspended matter (TSM) [23,24]. Different wavelengths of light penetrate water bodies to different extents. For instance, blue light can penetrate deeper into clear waters than green or red light. Conversely, in more turbid waters, red and near-infrared (NIR) light is absorbed more rapidly and scatters quickly, leading to a diminished signal at the surface. Hence, the spectral signatures captured by remote sensors are significantly influenced by the composition and clarity of the water body.

Various algorithms exist to calculate water parameters from multispectral data, including mechanistic models [25], artificial neural networks [26], and regression algorithms [27]. Each offers unique advantages and applicability, depending on the specific characteristics of the data and the aquatic ecosystem under study. For instance, a study by Chusnah and Chu [28] demonstrated the application of machine learning in estimating Chl-a concentrations, which are commonly used as indicators for assessing the trophic level of lakes and the state of water quality. The study utilized machine learning to implement a band ratio algorithm and generate Chl-a maps from Sentinel-2 and Sentinel-3 satellite images. However, mechanistic models, relying on physical laws and principles to simulate light attenuation and scattering processes in water, often provide the most reliable and accurate predictions, particularly when dealing with smaller datasets, which would not be sufficient for machine learning models [29].

Lee et al. [30] provided a foundational understanding of the optical properties influencing SD, which significantly contributed to the development of the quasi-analytical algorithm (QAA) for more accurate and reliable water clarity estimations. The QAA, a commonly used mechanistic model for SD, has been utilized in various water bodies [29,31,32]. It has been applied to MODIS and MERIS satellite data, where it reduced the root-mean-square error (RMSE) of SD estimation from 1.5 m to 1.0 m. Furthermore, the QAA has been used to account for the residual error in reflectance data from MODIS satellite data, demonstrating its potential for remote sensing in monitoring and managing water resources [33]. The algorithm showed excellent results ($R^2$ = 0.96, MAPD = 0.18) when validated with independent measurements covering oceanic, coastal, and lake waters [34]. However, previous studies have not delved into the potential benefits of utilizing multispectral cameras onboard UAVs in combination with the QAA algorithm. The integration of these cameras with UAVs offers potential improvements in spatial resolution and data availability, bridging the gap between in situ and satellite remote sensing measurements [18].

It is important to accurately account for reflected light from the water surface—more specifically, the sun-glint effect—as it can lead to inaccuracies in further processing of water quality algorithms [18,35,36]. The simplest way of avoiding sun glint is careful UAV flight time and direction planning; however, since the water surface is often uneven, it is hard to reduce sun glint completely [37]. There are several methods of reducing the sun-glint effect in multispectral UAV images during postprocessing, for example, M Muslim et al. [36] tested four methods proposed by Lyzenga et al. [38], Joyce [39], Hedley et al. [40], and Goodman et al. [41] and applied them to either the whole image or just the glinted area.

Other studies used the methods of Hochberg et al. [42], removing glinted pixels as NIR threshold or using HydroLight simulations [18]. The application of these methods should align with the specific requirements of the study. For instance, if the primary objective entails bottom mapping, the method proposed by Lyzenga offers superior results, as demonstrated in M Muslim's et al. [36] study. On the other hand, if the analysis is focused on assessing surface water quality parameters, the Hedley method emerged as the preferred choice by Windle and Silsbe [18]. Thus, the selection of sun-glint correction techniques requires careful consideration of the objectives, ensuring the most effective and accurate outcomes in different scenarios.

By yielding accurate, timely, and spatially inclusive data on water transparency, UAVs equip decision-makers with invaluable resources, necessitating immediate intervention or remediation measures. This is particularly evident when we consider the capabilities of UAVs for mapping vast areas. A single UAV flight, which takes approximately 25 min, can effectively map an area as large as 25 ha.

This research aimed to evaluate the effectiveness of QAA for the determination of SD using multispectral cameras onboard UAVs. It was performed by testing QAA on image datasets preprocessed using five different methods of sun-glint correction. Additionally, the study examined how water constituents and solar zenith angle affected the discrepancy between actual and modeled SD values. We hypothesize that the radiometric accuracy of a calibrated UAS sensor should meet the required accuracy of 5%, expected with ocean color remote sensing, when compared to in situ measures of Hooker et al. [43] and application of sun-glint correction methods will improve usability in SD modeling, by deviation of RMSD of the same 5% from in situ measures, which would still provide practical results for water quality assessment.

## 2. Materials and Methods

The study was performed in 43 water bodies in Lithuania (Figure 1), represented by high variability and proportion of optically active in-water components—turbidity, Chl-a and CDOM (Supplementary Table S1)—thus bolstering the robustness and generalizability of the findings. The research was conducted from May 2021 to May 2023.

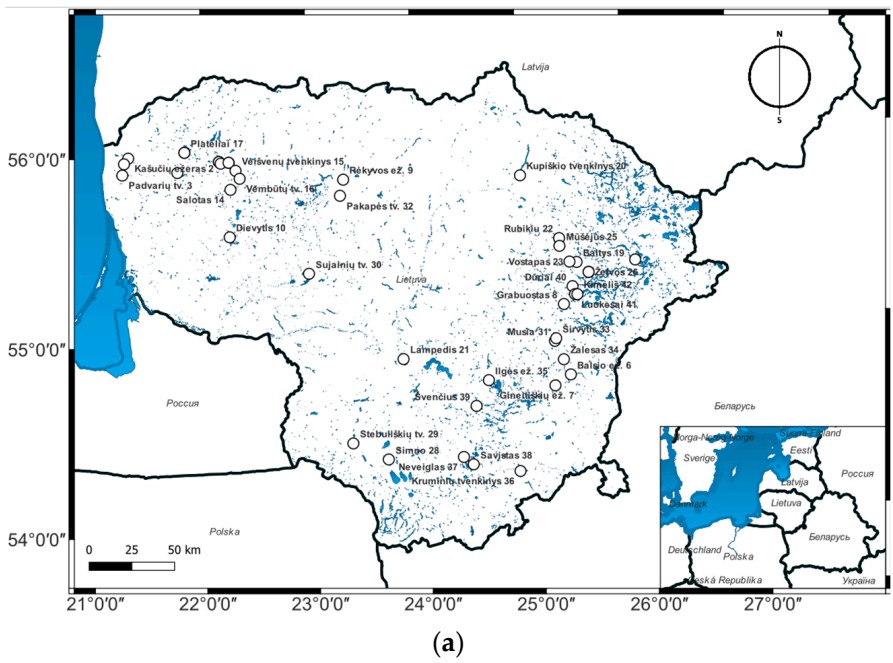

(a)

**Figure 1.** *Cont.*

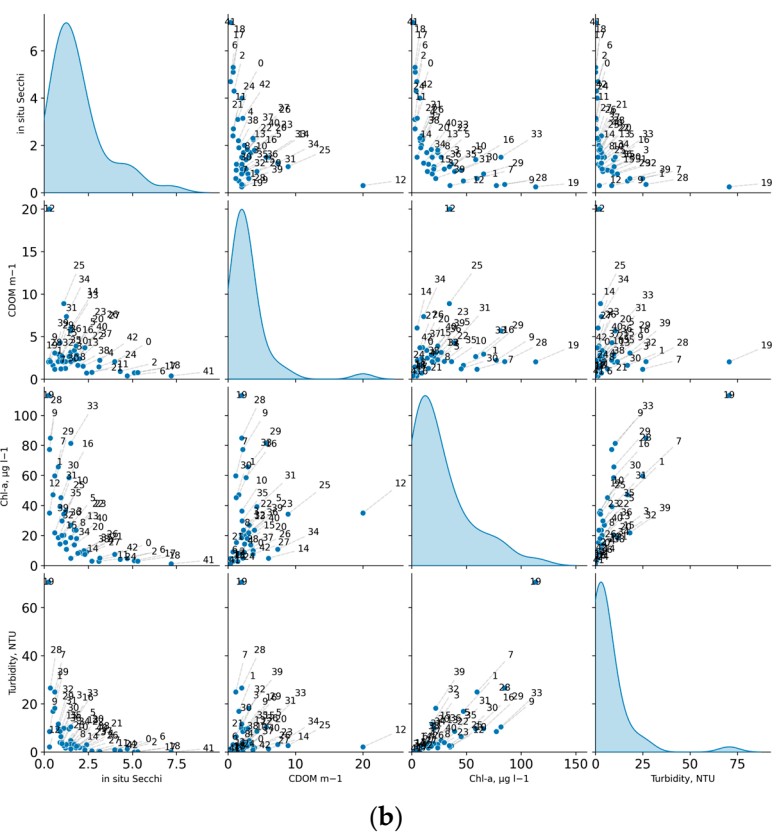

**(b)**

**Figure 1.** (**a**) Study area with indicated study sites. (**b**) Pair plot between in situ parameters, measured in the areas of interest. The diagonal plots show kernel density estimations for each parameter, offering a smoothed representation of the distribution of data values. ID numbers in pair plots represent the lake number after the name in the map with *n* = 43.

### 2.1. UAV Data Collection and Calibration

An Inspire 2 UAV equipped with a RedEdge-MX camera was used to facilitate the acquisition of data [44]. The camera features five bands: blue (475 nm ± 16 nm), green (560 nm ± 13 nm), red (668 nm ± 8 nm), red edge (717 nm ± 6 nm), and NIR (842 nm ± 28 nm). To ensure optimal image capture, the UAV was flown at a height of 60 m to optimize the balance between spatial resolution and the area covered in each image, allowing for both relatively high-resolution imagery (from 3 to 4 cm/pixel) and a reasonable area coverage per flight (around 40–50 m$^2$), with the camera programmed to capture images every three seconds. This frequency was necessitated by the inability to view the live feed and capture images in desired areas from the Micasense RedEdge-MX camera at a distance. The maximum distance between the SD measurement site, where GPS coordinates were recorded, and the image GPS coordinates were approximately 40 m, while the minimum distance was 0 when the measurement was above the boat. For the majority of water bodies, a single measurement site was selected in alignment with the methods of the Environmental Protection Agency of Lithuania. However, for a few larger water bodies, two sampling points were obtained for a more comprehensive understanding. The operation of the UAV was coordinated by two individuals: one controlling the UAV from the shore and the other onboard a boat, both communicating via phone during the Secchi disk measurements. The onboard operator also concurrently measured other water quality parameters outlined in Section 2.2.

For image calibration, the original Micasense RedEdge calibration panel was used. In accordance with Micasense's guidelines, a calibration panel image was captured at the shore before and after each flight, ensuring no shadows were cast on the reference panel or drone [44]. Three images were taken each time for assurance. Furthermore, the

Downwelling Light Sensor (DLS2) provides real-time, continuous measurements of the ambient light conditions during the flight, ensuring that the images captured are properly calibrated regardless of changes in the lighting environment.

## 2.2. In Situ Data

Concurrently with the UAV image capturing, in situ SD measurements were performed using a 30 cm white Secchi disk, a widely accepted method for assessing water transparency in lakes, rivers, and oceans [45].

The parameters of CDOM—the light-absorbing component of dissolved organic matter, Chl-a (an indicator of phytoplankton biomass), and turbidity (a measure of the cloudiness or haziness of water)—were measured alongside SD to assess their potential influence on water transparency. Parameters were only measured in areas where SD was lower than the depth of the specific point being assessed.

Water samples for Chl-a measurements were filtered through glass fiber GF/F filters with a nominal pore size of 0.7 μm and extracted into 90% acetone. Photosynthetic pigments were measured spectrophotometrically and estimated according to the trichromatic method [46,47]. CDOM was measured spectrophotometrically after filtration through 0.22 μm membrane filters. The CDOM absorption coefficient at 440 nm was derived according to Kirk [48]. A Shimadzu UV-2600 spectrophotometer was used for the analysis of Chl-a and CDOM. Turbidity was measured with a portable turbidity meter (Eutech InstrumentsTN-100, Landsmeer, The Netherlands) in the Nephelometric Turbidity Unit (NTU). The instrument has a light-emitting diode in the near-infrared range (Hach Lange at 860 nm and Eutech Instruments at 850 nm), and the detector measures the scatter at a 90° angle. This method is based on International Organization for Standardization (ISO) 7027.

In situ remote sensing reflectance $R_{rs}$ was acquired to validate wavelengths from UAV observations of water surface reflectance. $R_{rs}$ was measured in the spectral range of 400–800 nm by simultaneous measurements of downwelling irradiance, upwelling radiance, and downwelling radiance, performed with a WISP-3 spectroradiometer [49]. $R_{rs}$ was calculated according to Equation (1):

$$R_{rs} = \frac{L_u - \rho L_d}{E_d}. \tag{1}$$

where $L_u$ is the upwelling radiance, $L_d$ the downwelling radiance, $E_d$ the downwelling irradiance, and $\rho$ a water surface reflectance factor equal to 0.028.

Central wavelengths of the Rededge MX camera were used (475 nm, 560 nm, 668 nm, 717 nm, 842 nm). NIR data were excluded from comparison with in situ reflectance, which was not measured at 842 nm; however, NIR was still used for the sun-glint correction step Section 2.4.

## 2.3. Preprocessing the UAV Data

The collected UAV images underwent preprocessing of correcting atmospheric effects and standardizing the format suitable for analysis (Figure 2). The conversion of raw images to radiance involves several corrective steps to eliminate biases and errors, accounting for dark-pixel offset, vignette effect, as well as aligning images due to distances between sensors, which could potentially affect the accuracy and reliability of the data. These steps were achieved using the Micasense Python (version 3.7, Python Software Foundation, 2018) workflow, mainly the function raw_image_to_radiance, as described in the Micasense Github repository for users [44].

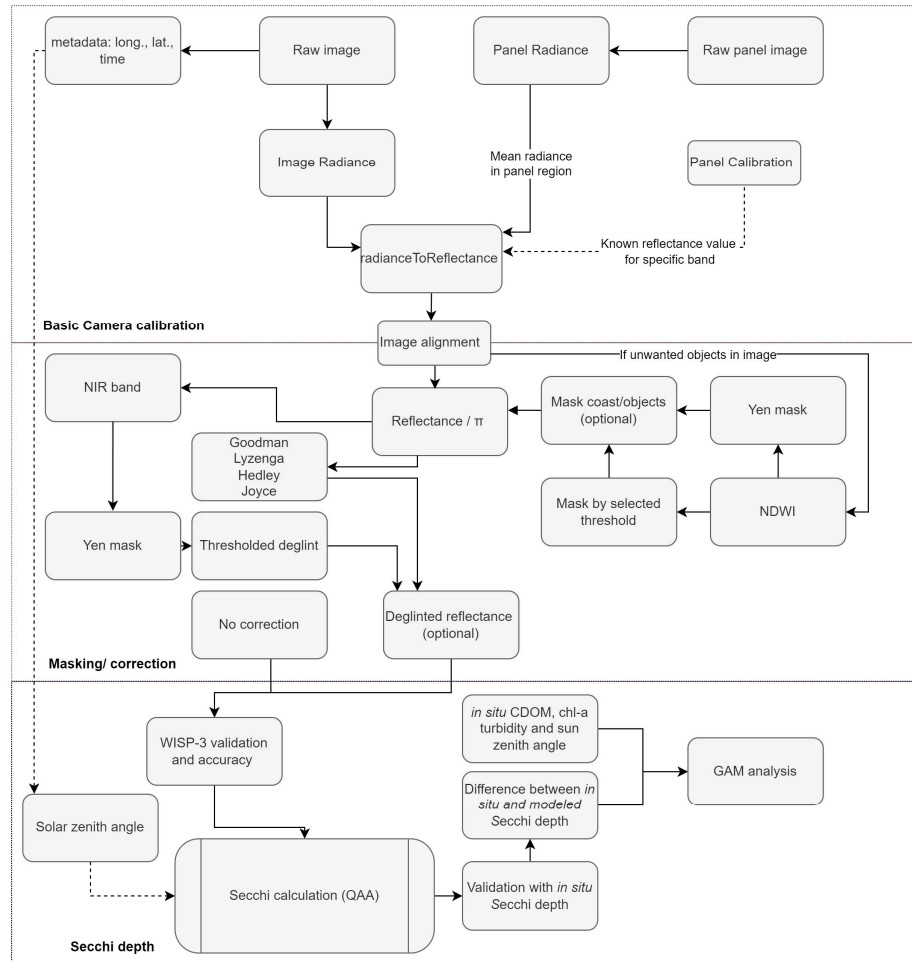

**Figure 2.** Processing workflow for selected UAV images. Arrows represent image processing and the dashed line arrows represent metadata. NDWI, normalized difference water index.

Since reflectance is relatively independent of illumination conditions [11,50], it is preferred for calculating SD using remote sensing images. The conversion of radiance to reflectance was performed using a reference panel of known reflectance to determine a scale factor between radiance and reflectance. This scale factor was applied to the entire image to obtain a reflectance image. The accuracy of the reflectance image was verified by extracting and checking the same reflectance panel region used to calculate the scale factor for any trends or inconsistencies [44].

Reflectance was further normalized by dividing it by $\pi$, assuming water as a Lambertian body, following the mathematical model (2).

$$\text{Reflected radiance} = \rho \times (\cos \theta\_i)/\pi \qquad (2)$$

Here, $\cos \theta\_i$ represents the cosine of the angle between the incident light and the surface normal and $\rho$–a surface reflectance factor [50]. This normalization ensures that the reflectance values fall within a standardized range (0 to 1), facilitating consistent comparisons and calculations across different surfaces, lighting conditions, and measurement devices [51,52].

### 2.4. Image Masking and Sun-Glint Correction

From the initial set of water bodies, four were eliminated (Tūbausių Reservoir, Grabuostas Lake, Ilgis Lake, and Mušėjus Lake) from further analysis. One was excluded due to excessive cloud glint, while the other three measurements were performed too close to the shore or contained emerged macrophytes, leaving no sufficient area where the reflectance

would not be affected (Figure 3a). One water body (Kruminių Reservoir) was surveyed at late in the day, when the zenith angle was approximately 80 degrees, resulting in shadows covering half the area; however, the unshaded area was able to be used for the analyses (Figure 3b), removing the shadowed area using binary thresholding. In total, 39 water bodies were left for statistical analysis after discarding unsuitable ones.

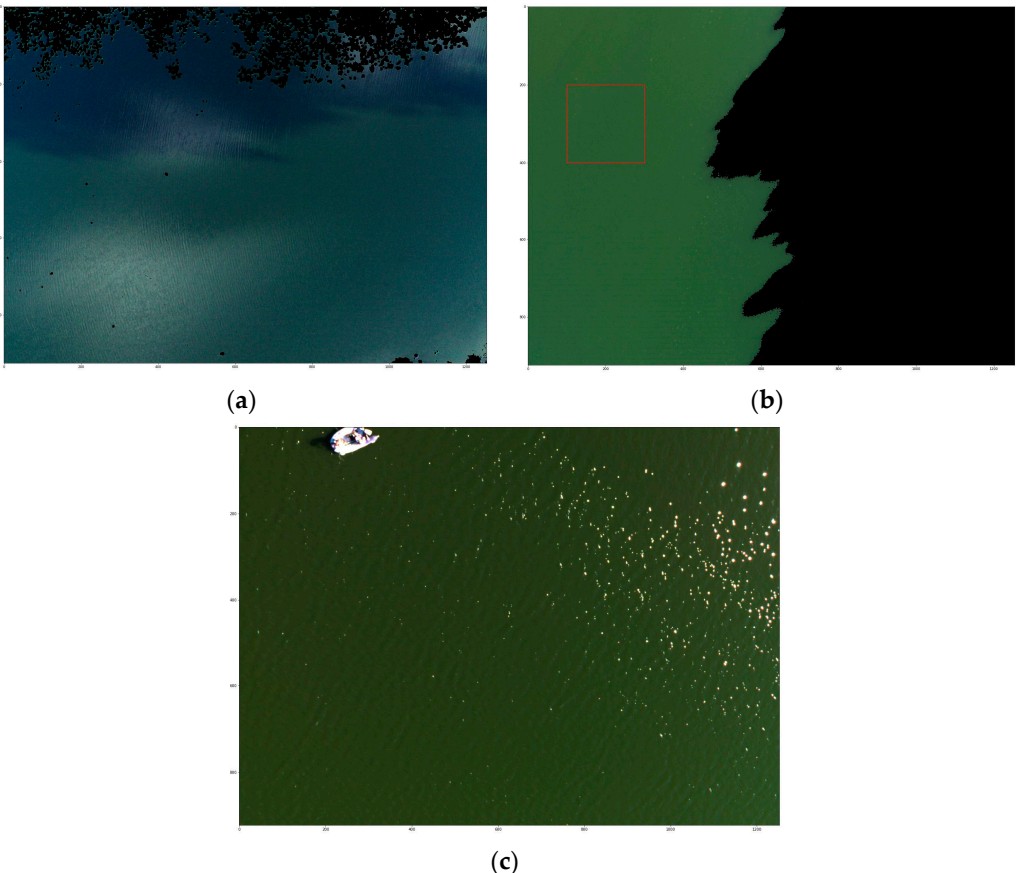

**Figure 3.** (**a**) RGB of an image affected by shadows, bottom reflectance and floating macrophytes after thresholding (Grabuostas Lake), (**b**) RGB of contaminated image (Kruminių Reservoir) with shadow and shore, but where values can still be used by selecting an area (red square) that was good, (**c**) RGB example of a relatively good image (Musia Lake) before removing sun glint and boat.

To ensure accurate SD measurements, objects (boats, coast, macrophytes) and shadows potentially affecting the measurements' accuracy were removed. Firstly, the normalized difference water index (NDWI) was calculated to distinguish between water and non-water pixels (3). This index was then classified into binary parts using Yen thresholding [53,54], where non-water pixels were masked as NaN values. Yen method was used also on just the NIR band to classify areas where pixels were affected by sun glint and masked as NaN in one of the sun-glint removal tests. This sun-glint removal method was later called the threshold-removed glint method.

$$NDWI = \frac{RrsGreen - RrsNir}{RrsGreen + RrsNir} \qquad (3)$$

In 11 images, water waves caused a substantial amount of sun glint (Figure 4a), despite images being captured early in the morning around 10 a.m. GMT+3, when the sun reflection from calm water should not have reached the lens. Therefore, the sun-glint correction was performed using the algorithms developed by Hedley et al. [40], Goodman et al. [41], Lyzenga et al. [38], and Joyce [39], which assume that the water surface reflectance is a

linear combination of water reflectance and sun-glint reflectance. The models were fitted using a set of training data consisting of image pixels where sun glint was absent, calculated from an area with 10% lowest value NIR pixels [18]. The model was then used to predict water reflectance for sun-glint-affected pixels, reducing reflectance values according to Equations (4)–(7).

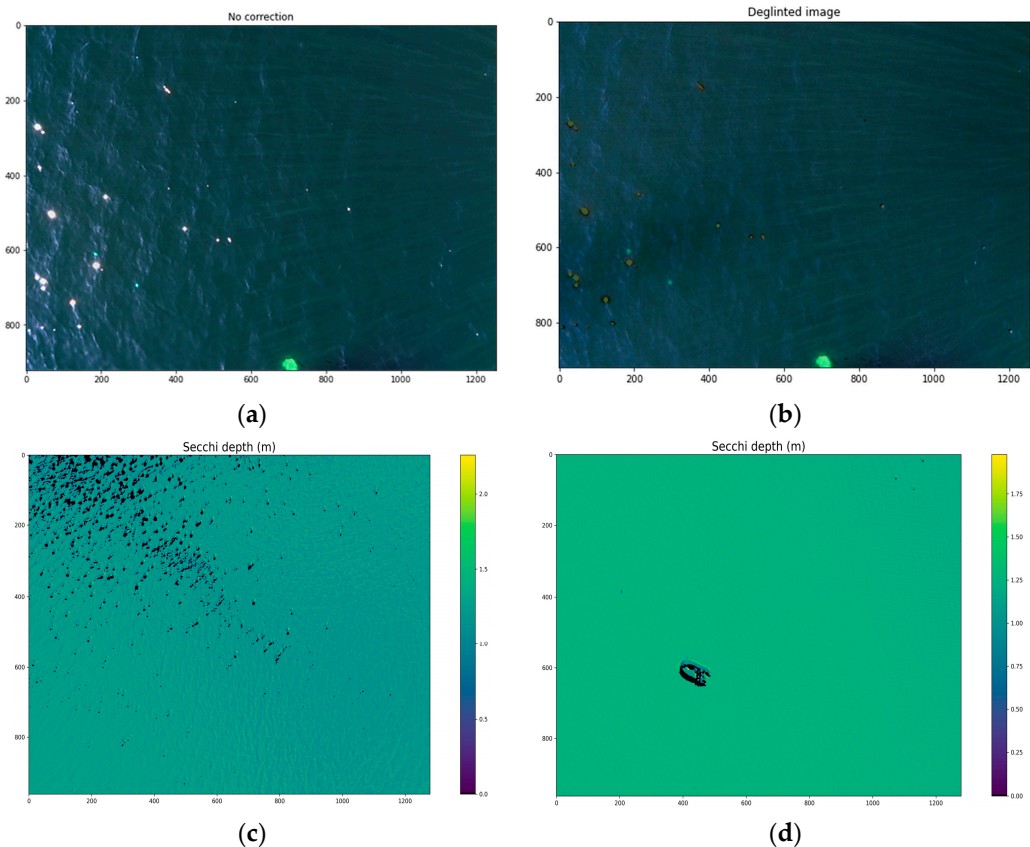

**Figure 4.** (**a**) RGB image before the sun-glint correction. (**b**) Deglinted RGB image using Hedley's method. (**c**) Threshold-removed sun-glint area from NDWI image, recalculated to SD; black areas represent removed values. (**d**) The boat removed from NDWI using Yen's threshold and recalculated to SD; black areas represent removed values.

The Hedley method (4) calculates the remote sensing reflectance ($R_{rs}$) for each pixel in each band. The method first finds the minimum NIR value and then calculates the slope for each band using linear regression. The Lyzenga and Joyce methods ((5) and (6)) are similar to the Hedley method, but use the mean and mode, respectively, of the lowest 10% of NIR values instead of the minimum NIR value. The Goodman method (7) calculates the Rrs for each pixel in each band using a constant A (0.000019) and a factor B (0.1) that is multiplied by the difference between the red and NIR bands:

$$\text{Hedley } R_{rs\lambda} = \lambda + bi_\lambda \times (NIR - NIR_{min}) \tag{4}$$

$$\text{Lyzenga } R_{rs\lambda} = \lambda + bij_\lambda \times (NIR - NIR_{mean}) \tag{5}$$

$$\text{Joyce } R_{rs\lambda} = \lambda + bi_\lambda \times (NIR - NIR_{mode}) \tag{6}$$

$$\text{Goodman } R_{rs\lambda} = \lambda - NIR + \delta, \text{ where } \delta = A + B \times (Red\lambda - NIR\lambda) \tag{7}$$

where λ is the band of interest (blue, green, red, red edge, NIR), $bi_\lambda$ is the slope of the band, $bij_\lambda$ is equal to covariance between λ and NIR divided by variance in NIR, and $NIR_{min}$, $NIR_{mode}$, and $NIR_{mean}$ are the minimum, mode, and mean NIR values, respectively. For Goodman's method, δ is a constant offset across all wavelengths where A and B are constants (A = 0.000019 and B = 0.1). Additionally, removing only sun-glint-affected areas (determined by the binary thresholding Yen algorithm) was compared as an alternative approach to sun-glint correction methods (Figure 4c).

The images of four water bodies (Vembutų, Stebuliškių, Pakapės, and Krūminių reservoirs) were also affected by either shore or cloud-glint artefacts. However, most of these artefacts were successfully removed using binary thresholding, except for one image (Figure 3b) where the area had to be manually chosen because the thresholding method was unable to accurately separate the unwanted areas.

### 2.5. Secchi Depth Model

The quasi-analytical algorithm (QAA) proposed by Lee et al. [30] provides a robust framework for monitoring SD (8), particularly in scenarios where in situ measurements may be unavailable. Compared to the empirical approach, this semianalytical method offers a significant advantage: it does not necessitate the recalibration of the retrieval algorithm with in situ data [31]. This enhances its utility in diverse settings. As such, it was considered more suitable for monitoring SD in various water bodies.

Lee et al. [30] introduced a mechanistic model that accounts for the effects of light attenuation, scattering, and reflection in the water column, as well as the properties of the Secchi disk itself, in determining the SD. The parameters that determine SD in this algorithm are the total absorption coefficient '$a$ and the total backscattering coefficient '$bb$. From these parameters, the diffuse attenuation coefficient $K_d$ was calculated, which is the main variable in the SD formula, besides the $R_{rs}$ band:

$$K_T\_K_d = \frac{1.04 * (1 + 5.4u)^{0.5}}{1/\left(1 - \frac{sin(\theta)^2}{RI^2}\right)^{0.5}}.$$  (8)

$$SD = \frac{1}{kt\_kd * minKd} \, ln\left(\frac{0.14 - minRrs}{0.013}\right)$$  (9)

where minKd is the minimum value of $K_d$ chosen from $K_d$ calculated with blue, green and red bands and minRrs is the above-surface remote sensing reflectance of the band that had the lowest $K_d$ value. $K_t\_K_d$ is the upwelling radiance diffuse attenuation coefficient and was used instead of a constant value of 1.5 as suggested by Jiang et al. [31], where θ is the solar zenith angle, *RI* is the refractive index value of water equal to 1.33 and *u* denotes the ratio of backscattering coefficient to the sum of absorption and backscattering coefficient.

The sun zenith angle was computed from in situ measurements using the time and location of the observation. This information was then used to calculate the solar position using the Python library pytz [55]. This approach provided a reliable means of determining the solar zenith angle, which is an important parameter in applications related to SD acquisition from $R_{rs}$.

Selecting the appropriate reference wavelength is of major importance for the final SD value [34]. The green band (560 nm) was used as a reference if the red band $R_{rs}$ (668 nm) was <0.0015 sr$^{-1}$. Otherwise, the red band (668 nm) was used, with accordingly modified calculations of other parameters, as suggested by Lee et al [30].

SD was calculated using the QAA method for images of 6 types: corrected after Hedley, Lyzenga, Joyce and Goodman sun-glint algorithms, threshold-removed images and images with no correction.

*2.6. Validation and Interpretation of Results*

All of the final modeled SD values and reflectance images were averaged to ensure that images could be compared with in situ data. Pearson correlation coefficients were calculated to assess multicollinearity between the reflectance values of different wavelengths. The final reflectances obtained after applying various correction methods and no correction (Sections 2.3 and 2.4) were validated with in situ reflectances (WISP-3). The accuracy of these methods was evaluated by bias (9), while the root-mean-square deviation (RMSD) was used as an indicator of the QAA model's precision (10), and the Pearson's correlation coefficient (r) described the relationship strength between the model's output and the real data values.

$$\text{Bias} = \frac{1}{N}\sum_{i=1}^{N}(X_{estimated,i} - X_{measured,i}) \tag{10}$$

$$\text{RMSD} = \sqrt{\frac{\sum_{i=1}^{N}(X_{estimated,i} - X_{measured,i})^2}{N}} \tag{11}$$

The same accuracy and precision measures were applied to modeled SD values. Generalized additive models (GAMs) were employed [56] to investigate the relationships between the difference in modeled and in situ SD (the response variable) and a set of independent variables: CDOM, Chl-a, turbidity, and solar zenith. The GAMs were chosen due to their flexibility in modeling nonlinear relationships and their ability to handle interactions between predictors. The GAMs were utilized using R programing language with the mgcv [57] library for statistical parameters and ggplot2 [58] library for visualization. Before analysis, the cross-correlation (based on the Pearson correlation coefficient) between the independent variables was determined. The correlation was relatively high (r = 0.76) between Chl-a and turbidity; therefore, turbidity was not included in the GAMs. Before the interpretation of the GAM results, the residuals were visually inspected with diagnostic plots for normality and equal variance against the fitted values. F and *p* values were obtained to assess the relative importance and significance of the independent variables (*p* < 0.05 was considered a statistically significant relationship). The fit of the model was evaluated using the explained deviance. A response plot was graphically represented to visualize the relationship between an explanatory variable and the response. As there were multiple predictors, each one was plotted separately with a smooth curve and the confidence interval of the effect.

To visualize the SD results, a subset of images with calculated SD values were exported as TIFF and mosaiced according to image metadata GPS coordinates in QGIS version 3.16 [59]. The result was then visualized on top of the RGB mosaic that was mosaiced using OpenDroneMap [60] photogrammetry software.

## 3. Results

*3.1. Band Validation after Sun-Glint Correction*

Across all correction methods and the in situ data, the green band (560 nm) consistently showed the peak mean value, while the blue band (475 nm) indicated the valley or lowest mean value. The general shape of the data appears to peak at the green band, with decreasing values on either side at the blue and red bands (668 nm), and a slight increase at the red edge band (717 nm). This pattern was consistent across all the correction methods and the in situ data, indicating the robustness of this spectral feature in the multispectral UAV image data. In terms of multicollinearity, the green and red exhibited the strongest correlation of 0.97, closely followed by correlations between the green and red edge at 0.96 and the red and red edge at 0.95. The blue band also exhibited strong multicollinearity with the other bands: 0.97 with green, 0.94 with red and 0.91 with red edge.

The highest mean values across all bands were observed with the Lyzenga correction method, while the lowest was observed with the Goodman correction method (Figure 5). The in situ values were generally lower than the corrected values, except for the Goodman correction method.

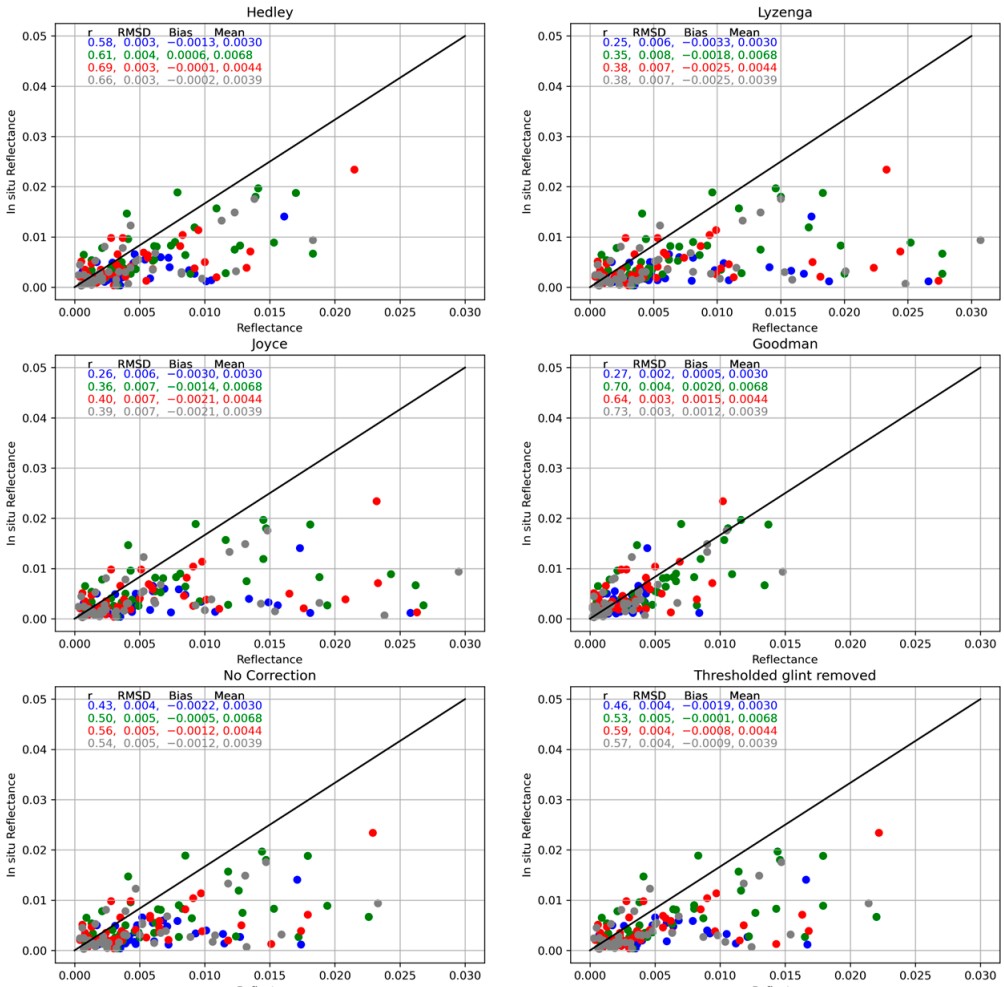

**Figure 5.** Spectral scatterplots demonstrating the agreement between in situ measured and sun-glint-corrected (or not corrected) reflectances in the UAV images. Statistics (r, RMSD, bias) and points are color-coded accordingly, except for the red edge, which is gray. A diagonal black line marks perfect agreement (1:1).

Removing the two outlier points, one from green and one from the red edge band, decreases r values and the r mean becomes similar to the red bands. However, it also decreases RMSD and bias for these bands, thus not decreasing the accuracy of the data.

The validation results provided robust evidence regarding the performance of the correction methods and the reliability of the final reflectance values. The correlation for all methods (Figure 6) had the same trend, where it was the lowest for the blue band (r = 0.38 ± 0.12) as well as for the red band (r = 0.56 ± 0.14), but higher for the green band (r = 0.75 ± 0.11) and red edge band (r = 0.80 ± 0.10). However, RMSD was relatively low for the blue band (RMSD = 0.0043 ± 0.0016) and red (RMSD = 0.0048 ± 0.0018) bands and slightly higher for the green (RMSD = 0.0060 ± 0.0013) and red edge (RMSD = 0.0053 ± 0.0014) bands. Bias for most methods followed a similar trend of larger underestimation for the blue band and green bands, then slight overestimation for the red band with Goodman, Hedley and threshold-removed glint methods, but still undervaluation for other methods, and slightly lower undervaluation for the red edge band. Goodman's method was exceptional to these trends, as green, red and red edge bands were overestimated.

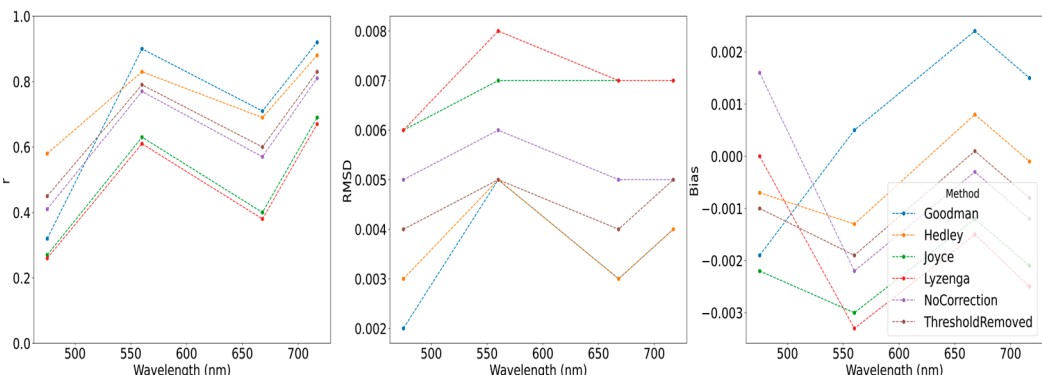

**Figure 6.** Mean r, RMSD and bias between corrected reflectances with each method and in situ measured reflectances across all available wavelengths. RMSD and bias are presented in reflectance values.

Overall, Goodman's algorithm showed the highest correlation for the green and red edge bands, with r values of 0.90 and 0.92, respectively. However, it had low r values for the blue and red bands, with 0.32 and 0.71, respectively. The RMSD values for Goodman were relatively low across all bands, ranging from 0.002 to 0.005. Similarly, the Hedley algorithm consistently performed well across all bands, with r values ranging from 0.58 to 0.88. The RMSD values were similar to those of Goodman, ranging from 0.003 to 0.005.

The Joyce and Lyzenga algorithms showed matching performances, with lower r values and higher RMSD and bias values compared to Goodman and Hedley. The r values ranged from 0.27 to 0.69 for Joyce and from 0.26 to 0.67 for Lyzenga. The RMSD values ranged from 0.006 to 0.007 for both algorithms.

The control group with no correction applied showed moderate r values, ranging from 0.41 to 0.81. The RMSD values were similar to those of Goodman and Hedley, ranging from 0.005 to 0.006.

### 3.2. Validation of QAA SD Model

The performance of the QAA model's (Figure 7) ability to predict SD when compared with in situ measurements showed a relatively high correlation across all methods, with r ranging from 0.74 (threshold-removed glint) to 0.92 (Hedley glint correction). RMSD (from 0.65 to 1.05 m) and bias (from −0.78 to 0.58 m) showed acceptable results for all methods as well.

Comparing methods between themselves, the accuracy of all parameters had similar trends as accuracy for band comparison with in situ measurements (Section 2.1), where the SD values were overestimated for the smaller SD values and underestimated for the larger ones, except for Goodman's method, where most of the SD values were overestimated.

Hedley's sun-glint-corrected images achieved the best results according to the RMSD measures (0.65 m), while r was just slightly smaller (0.91) than with Goodman's method (0.92), which achieved the best results based on the r value of 0.92. However, the RMSD value for Goodman's method was relatively high (1.00 m) compared to the values of other methods. This method overestimated most of the values for both small and large SD values.

The Joyce and Lyzenga methods showed similar results between themselves with r values of 0.87 and 0.85, respectively. The RMSD values for Joyce and Lyzenga were 0.79 and 0.86 m, respectively, and biases showed underestimation.

The worst-performing method was when only sun-glint-affected pixels were removed and all other pixels were left unchanged (r = 0.75, RMDS = 1.05 m, bias = 0.13 m). The control group with no correction showed slightly better results: r value of 0.89, RMSD value of 0.74 m and underestimation according to bias.

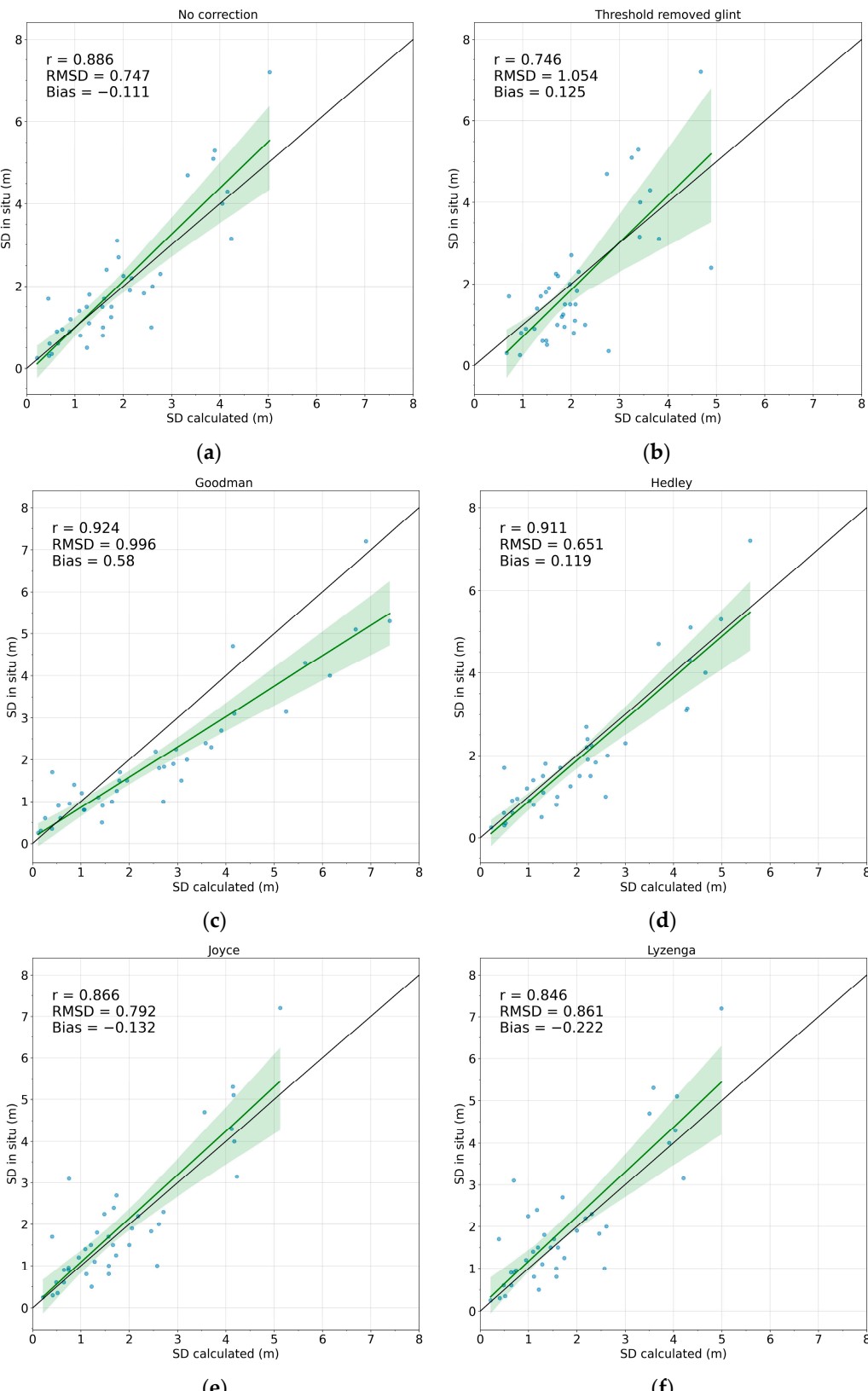

**Figure 7.** Scatterplots with linear regression line and 95% degrees of freedom (green) of modeled and in situ SD. Best-fit line (1:1) (black) for each image dataset, using different preprocessing methods: (**a**) no correction, (**b**) threshold-removed glint, (**c**) Goodman, (**d**) Hedley, (**e**) Joyce, (**f**) Lyzenga ($n$ = 39).

### 3.3. Relation with Water Constituents

The average in situ SD of the lakes was 1.91 ± 1.53 m (±standard deviation), ranging from a minimum of 0.25 to a maximum of 7.2 m. The mean CDOM was 2.95 ± 3.2 m$^{-1}$ (min–max: 0.37–20.01 m$^{-1}$), the mean Chl-a concentration 26.59 ± 26.48 (1.13–113.23) and the mean turbidity 6.76 ± 11.8 NTU (0.00–70.62 NTU).

The environmental factors in GAMs significantly explained (38.3%) the variance in the difference in SD measurements. The interaction term of the sun zenith angle and CDOM was significant (F = 6.808, *p*-value < 0.05), suggesting that these factors together affect the accuracy of SD retrieval (Figure 8). The most important and statistically significant factor was CDOM (F = 10.47, *p* < 0.05), followed by the solar zenith (F = 4.84, *p* = 0.02). Chl-a values did not have a significant effect on the GAM model (F = 0.295, *p* = 0.59).

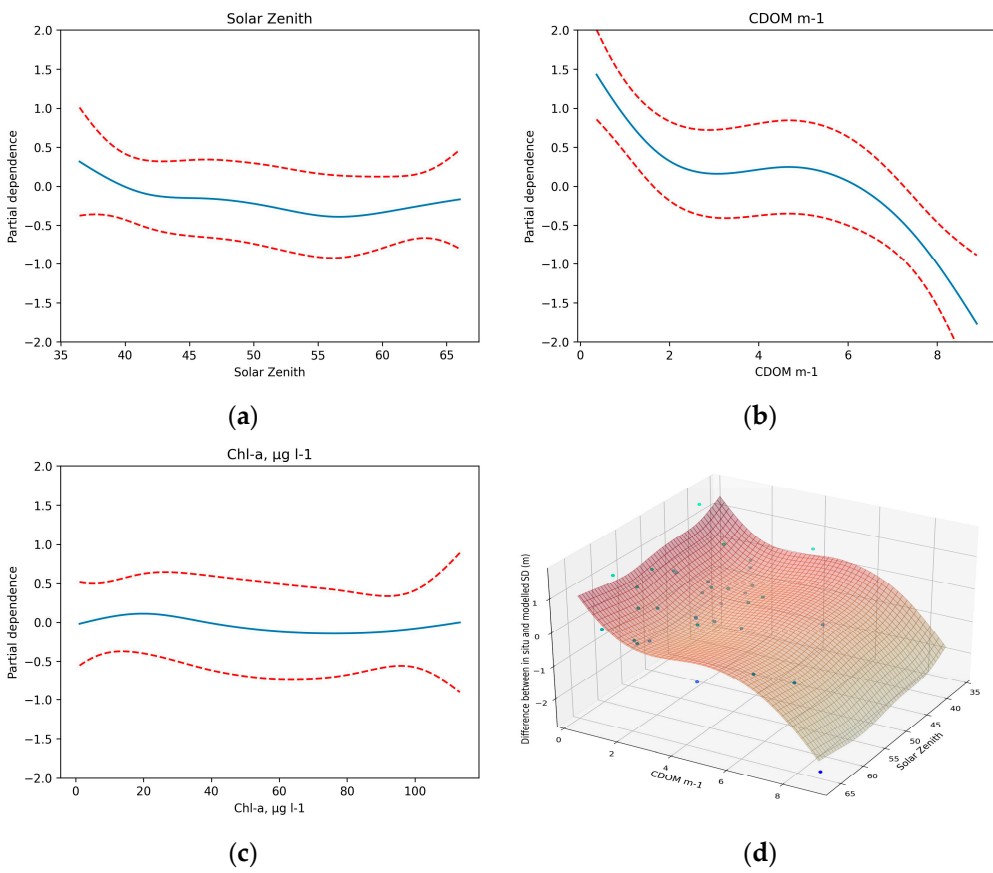

**Figure 8.** GAMs fitted smooth lines (blue lines) between the difference in SD (modeled vs. in situ) and graphs for each independent environmental factor: (**a**) solar zenith (**b**) CDOM (**c**) Chl-a and (**d**) interaction plot of solar zenith angle. CDOM and predicted difference plane between in situ and modeled SD (m). The red dashed lines show a 95% confidence interval for fitted lines.

In the residual plots of the model a random scatter of points was observed (Figure 9), with no discernible pattern or trend. This scatter indicates that the residuals have a constant variance, which suggests homoscedasticity. The absence of any systematic structure or pattern in the residuals reinforces the notion that nonlinear relationships assumed by our model are an adequate representation of this dataset. Additionally, we did not identify any significant outliers that could unduly influence our model's predictions.

The modeled SD values were overestimated (to over 1.3 m) when CDOM values were <7.5 m$^{-1}$, while underestimated (over 1 m) when CDOM was >7.5 (although underestimation did not significantly change when CDOM > 12). For the solar zenith, the SD values were overestimated (over 0.7 m) at the lower solar zenith angles (<45 degrees), and then a relatively low effect (within ±0.5 m) was between 45 and 75 degrees of the solar zenith

angle and the underestimation sharply increased to over 1.5 m when the solar zenith angle was >75 degrees.

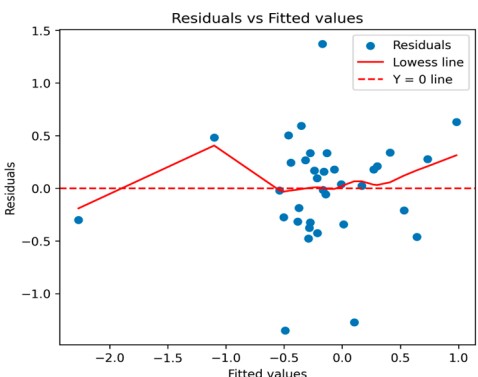

**Figure 9.** A scatterplot of residuals and fitted values with the red line presenting the best fit and dashed red line representing the Y = 0 line.

## 4. Discussion

### 4.1. Advancements in SD Measurements

This research focused on evaluating the effectiveness of the QAA for determining SD using a multispectral camera onboard UAVs. This novel approach expands the current understanding of measuring SD by introducing the potential use of UAVs, balancing the broad coverage provided by remote sensing methods with the high accuracy characteristic of in situ measurements, thereby serving as a more comprehensive alternative to traditional methods.

The experimental results did not uniformly support our hypothesis that the application of sun-glint correction methods would enhance the utility of multispectral images in SD modeling by reducing the RMSD by about 5% and reducing the bias by a similar amount. The precision observed across all bands appears to correspond with the spectral band reflectance intensity, where bands demonstrating relatively lower reflectance intensity (e.g., blue band) typically displayed a lower compliance between the reflectance measured from the UAV and the in situ reflectance measurements and the reverse held true for bands with relatively higher reflectance intensity. Environmental factors such as weak water surface signal and roughness of the water surface can introduce systematic and random errors, respectively, into water surface detection [13]. The correlation with the blue was identified as the least robust (Figure 7), which can be ascribed to increased vulnerability to scattering, a trait inherent to these bands in the water environment [61].

Among the tested glint correction methods, the Hedley and Goodman methods emerged as the most effective across all bands, with Hedley demonstrating the lowest RMSD across all bands. For instance, a study conducted by M Muslim et al. [36] employed a similar methodology to our study, testing multiple sun-glint correction methods. Their findings indicated that the Lyzenga method yielded the most accurate results. However, it is important to note that the primary objective of their research was to map coral reefs, and in most of the study area, the bottom was visible, which may have influenced their results. In contrast, Windle and Silsbe [18] found that the Hedley method provided the highest correlation coefficient (r) and the lowest RMSD, corresponding closely with the results of our study.

The key distinction between the three glint correction methods (excluding the Goodman method) lies in the slope or covariance index of NIR band values. In the case of the Hedley method, the slope was the largest, as it takes the minimum value of the band as the starting point, as opposed to the mode (Joyce method) or the mean (Lyzenga method) [38–40]. This suggests that the images used in our study required a larger numeric correction due to the lowest initial reflectance values.

The results showed that the Goodman's method reflectance intensity values tend to be significantly different from other methods for the blue, red and red edge bands. This method, which was ranked as the second-best-performing in our study, adopted a different approach, using constant values to correct for the sun-glint effect [41]. This approach led to overall better r values between in situ and the green, red, and red edge bands onboard the UAV. However, the performance of the blue band was significantly lower than when using the Hedley method. Despite this, the Goodman method demonstrated the lowest RMSD values and small overestimation bias, underscoring its effectiveness in applications where low reflectance intensity deviation from in situ radiometer is required.

The results also indicated that applying glint correction to the entire image rather than just the binary thresholded area can yield better results in terms of the overall accuracy of the reflectance values. This observation implies that the method of applying glint correction solely to the binary thresholded area might not be the most effective strategy, as it may overlook potential glint effects present beyond this area. Additional sun-glint correction research should be considered in the future for better generalization of reflectance correction [62].

The overall agreement between the in situ SD measurements and the modeled SD values might be connected to their handling of in-water constituents such as colored dissolved organic matter (CDOM) and solar zenith angles, which significantly influenced the accuracy of the models. CDOM was found to predominantly influence the discrepancies between in situ measurements and modeled SD. Given that CDOM primarily absorbs light in the ultraviolet (UV) and blue regions of the spectrum, resulting in a relatively low acquired signal by sensor [63], this agrees with our observation that the blue band demonstrated one of the weakest correlations between in situ and UAV-derived reflectance measurements [64]. It is plausible that Goodman's and Hedley's methods more effectively managed this factor, where Hedley's had the highest r and Goodman's had the lowest RMSD for blue band reflectance, resulting in a closer match with in situ values. Another potential interference is bottom reflection, particularly in clear waters, where the difference between the SD and the actual water depth is minimal. While the SD was consistently less than the water body depth in our study, situations where this difference is minimal could lead to bottom influences on the measurements.

Similarly, high solar zenith angles above 70° impacted the accuracy of modeled SD, leading to underestimations by up to 1.5 m. This is likely due to increased scattering and absorption of light at higher zenith angles, resulting in less light reaching the water's surface and thus larger differences between modeled and in situ SD values [65]. This reinforces the need for sophisticated algorithms that can accurately model these complex environmental factors in SD predictions.

### 4.2. Practical Applications

Accurate SD determination in large areas is particularly important given the increasing demand for high-resolution data on water transparency for applications including water resource management, environmental monitoring, and ecological modeling [66]. Measures of SD can reduce the need for boat measurements in lakes, also allowing for data collection at a higher frequency, surpassing traditional monthly monitoring intervals that may be inadequate for dynamic water bodies with recurring algal blooms, which in some cases can be inaccessible due to terrain or vegetation around the lake [67]. UAVs could also be used in shallow coastal waters (<1.5 m), where a research vessel usually cannot access them, e.g., in the Curonian Lagoon [68].

The optimal fit model using Hedley sun-glint correction on the whole image was used to construct a mosaic from 45 individual images (Figure 10) of Kašučių Lake on 20 September 2021. All of the pixels were left unmasked to show how shadows affect the final results, and therefore this aspect should be taken into consideration while planning the acquisitions.

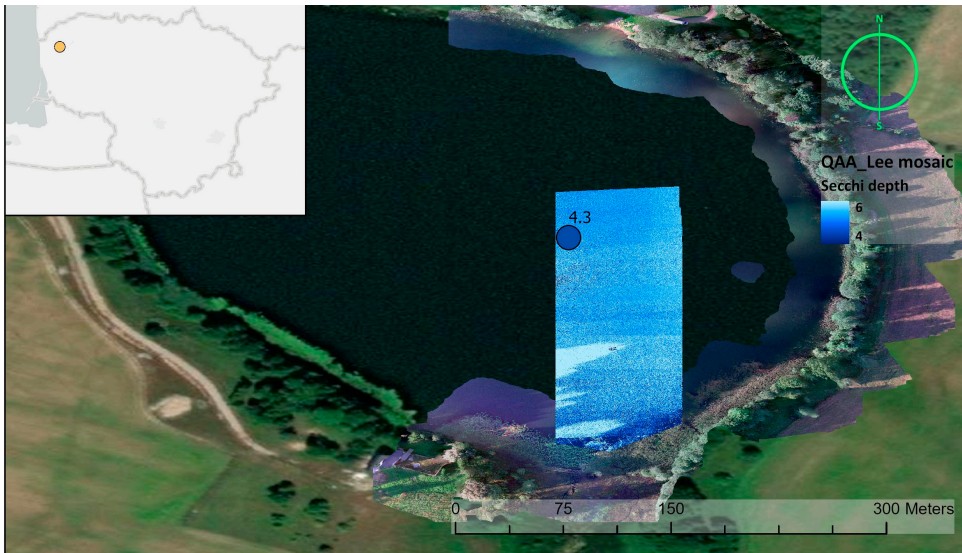

**Figure 10.** An SD transect of 45 mosaiced images in Kašučių Lake 20 September 2021. The dot with color represents in situ SD of 4.3 m.

This example shows the efficacy of the QAA best-fit model in transforming discrete data into a coherent, large-scale representation of SD. The SD varied from 4 to 6 m, where the highest values were determined in the areas covered by tree shadows. Elsewhere, SD values were more homogeneous (4.0 to 4.8 m), but some noise was still apparent in the orthomosaic (snow-like effect), as SD values are calculated for each pixel. For this reason, the mean value of the whole image was compared to in situ measurements in the area, instead of selecting a small square or point in the images, which would have resulted in lower accuracies.

This method has potential applications in shaping policies and regulations related to water bodies. With the methods provided in this paper, processing workflow for one 5 ha lake should not take longer than 20 min. Moreover, this process can be fully automated, requiring only the supervision of the final results. To put this into context, Lithuania has around 6000 lakes, and about 340 of them are larger than 5 ha [69], which would be suitable for monitoring using a Sentinel-2 satellite, assuming that the shape is not elongated. The rest of the smaller lakes could be monitored by applying methodology from this study and potentially could improve the accuracy and coverage of current national monitoring conducted by the Environmental Protection Agency, which currently covers around 80 lakes every 6 years in Lithuania.

### 4.3. Future Research and Potential Limitations

The QAA model possesses the flexibility to process remote sensing data sourced from an array of sensors, including but not limited to MODIS, MERIS, OLCI, MSI and GOCI. By integrating a more diverse set of global in situ measurements corresponding to various water types, it is conceivable to further refine the precision and effectiveness of the QAA model.

While the research results demonstrate potential, it is essential to emphasize that the investigation was conducted across a broad range of unique aquatic environments and included measurements captured under variable sun zenith angles. Consequently, for future work, there is potential to modify the existing QAA used for SD estimations to better account for CDOM and sun zenith angle, which in this study have been shown to be critical parameters. This modification could improve our understanding and predictions in the context of diverse and changeable aquatic conditions. Several limitations need to be addressed in future research. One of the main limitations is the assumption of linearity in the Hedley method. This method requires at least one dark pixel unaffected by glint [40] to

be visible in the image, and if the whole image is affected by glint, this deglinting method will provide incorrect data. Understanding these limitations aids in contextualizing the results and fosters the development of more refined glint correction methodologies in the future. This is also relevant for a partial cloud glint when clouds are reflected from the water surface in some, but not all areas of the image, for the analysis of multispectral drone imagery [40,70]. Most methodological studies on water surface mapping using UAVs suggest careful planning of flight time, preferably during conditions of clear skies and low sun glint [37]; however, this restricts one of the main advantages of UAVs—obtaining data on demand—especially if there is a need to visit several inland water bodies on the same day.

Many studies underscore the importance of choosing the appropriate quasi-analytical algorithm (QAA) for different optical water types [13,29,34]. The optical properties of coastal and inland waters, which are primarily influenced by the concentrations of suspended particulate matter, phytoplankton, and dissolved organic material, exhibit spatial and temporal variability, leading to diverse optical water types [71,72]. Applying a universal algorithm in these optically complex waters often results in significant uncertainties [71,72]. This insight highlights the importance of optical water classification to enhance retrieval accuracy, as indicated by numerous studies that developed class-specific algorithms for bio-optical parameters and achieved improvements by applying optical classification in the retrieval of these parameters [72–75]. In our study, relatively high concentrations of CDOM significantly affected the optical properties of the water, thereby introducing uncertainties in the results. One possible approach to improve the accuracy of the QAA is to calibrate the algorithm for the CDOM-dominated water bodies.

In addition to the aforementioned limitations, another significant challenge to consider is the interference of high-vegetation pixels in the analysis of multispectral drone imagery [76]. Vegetation and their shadows, especially when in close proximity to water bodies, can skew the reflectance measurements and thus influence the accuracy of Secchi depth estimates. Consequently, implementing strategies to exclude these pixels during image analysis can enhance the reliability of the measurements.

The applicability of water parameters extends beyond SD, allowing for the calculation of additional parameters. Prior research has demonstrated the feasibility of employing multispectral UAVs for turbidity [77–79], Chl-a [77,79–81], CDOM [79], TSS [80,82], cyanobacteria [80] and macrophytes [16,53,83]. Given the inherent scalability of the UAV-based methodology, it stands as a promising tool for extensive SD assessments and water quality surveys, thereby facilitating large-scale studies focused on water transparency. This advancement brings us a step closer to exploiting the full potential of UAV-based remote sensing for assessing and monitoring aquatic environments.

## 5. Conclusions

This study revealed that the accuracy of SD measurements is profoundly influenced by sun-glint correction methods employed in UAV flights. There was a consistent agreement across all methods and the in situ radiometric data, particularly for the green band, emphasizing the robustness of multispectral UAV image data. Among the tested methods, Hedley's method demonstrated superior accuracy (RMSD = 0.65 m) and precision, thereby significantly contributing to the accuracy of the UAV-derived SD data.

Moreover, findings underscored the significant role of environmental factors, particularly the CDOM and solar zenith angle, causing inaccuracies in SD measurements: a solar zenith angle > 70° resulted in an underestimation of up to 1.5 m in modeled SD, while CDOM > 12 $m^{-1}$ caused similar underestimations. Our research, therefore, supports the use of UAVs equipped with multispectral cameras as a viable method for SD determination in inland water bodies with SD of up to 7 m and lower than 12 $m^{-1}$ CDOM. The results point towards an approach capable of reaching a correlation as high as 0.91 and reducing the RMSD by up to 12.85% (Hedley's method), thereby enhancing the versatility and reliability of SD measurements.

**Supplementary Materials:** The following supporting information can be downloaded at https://www.mdpi.com/article/10.3390/drones7090546/s1. Table S1. Summary of investigated lakes and reservoirs in Lithuania. Descriptive statistics of in situ-measured SD, chlorophyll-a, CDOM and turbidity.

**Author Contributions:** E.T.: conceptualization, methodology, software, validation, formal analysis, investigation, resources, data curation, writing—original draft, writing—review and editing, visualization, supervision; M.B.: conceptualization, methodology, validation, investigation, resources, data curation, writing—review and editing, supervision; D.V.: conceptualization, validation, investigation, resources, data curation, writing—review and editing, supervision; J.G.: conceptualization, validation, investigation, resources, writing—review and editing; I.B.: resources, data curation, funding acquisition. All authors have read and agreed to the published version of the manuscript.

**Funding:** This research was supported by a doctorate scholarship program in Ecology and Environmental Sciences at Klaipeda University, Lithuania. The field campaigns were cofunded by the Environmental Protection Agency contract "Assessment of the water condition and its more efficient management of the remote monitoring data collection, processing, use and storage system, to ensure accurate results (NUOTOLIS)" (grant 28T-2021-64/SUT-21P-20).

**Data Availability Statement:** The data presented in this study are available on request from the corresponding author.

**Conflicts of Interest:** The authors declare no conflict of interest.

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
