# Peer review of "An Evaluation of Sun-Glint Correction Methods for UAV-Derived Secchi Depth Estimations in Inland Water Bodies"

_drones, doi:10.3390/drones7090546_

Round 1
Reviewer 1 Report
Please see the attached comments from the reviewer.

Well done. The reviewer found only minor issues with the manuscript.
Author Response
Dear reviewer,
Thank you for your insightful feedback on our manuscript. We've carefully considered your notes and made necessary revisions to improve our work. We hope our responses and the manuscript changes address your concerns effectively.
We appreciate your dedication to enhancing the quality of our paper.

Reviewer 2 Report
An Evaluation of Sun-Glint Correction Methods for UAV-De-2 rived Secchi Depth Estimations in Inland Water Bodies, Edvinas Tiškus *, Martynas Bučas, Diana Vaičiūtė, Jonas Gintauskas, Irma Babrauskienė
It’s quite an interesting study about how to estimate Secchi Depth from UAV multispectral imagery over 43 lakes in Lithuania. This manuscript is well described about strengths and limitations to collect drone-borne Secchi Depth measurements in the section of methods and discussions.
It would be better to put latitude and longitude coordinates in the figure 1. For example, the top of figure is placed for longitude while the left of figure is for latitude.
Suggesting relevant references for introduction and background sections in this manuscript as following:
Rowe, C. E., Figueira, W. F., Kelaher, B. P., Giles, A., Mamo, L. T., Ahyong, S. T., & Keable, S. J. (2022). Evaluating the effectiveness of drones for quantifying invasive upside-down jellyfish (Cassiopea sp.) in Lake Macquarie, Australia. Plos one, 17(1), e0262721.
Escobar-Sánchez, G., Markfort, G., Berghald, M., Ritzenhofen, L., & Schernewski, G. (2022). Aerial and underwater drones for marine litter monitoring in shallow coastal waters: factors influencing item detection and cost-efficiency. Environmental monitoring and assessment, 194(12), 863.
Brooks, C., Grimm, A., Marcarelli, A. M., Marion, N. P., Shuchman, R., & Sayers, M. (2022). Classification of Eurasian watermilfoil (Myriophyllum spicatum) using drone-enabled multispectral imagery analysis. Remote Sensing, 14(10), 2336.
Sibanda, M., Mutanga, O., Chimonyo, V. G., Clulow, A. D., Shoko, C., Mazvimavi, D., ... & Mabhaudhi, T. (2022). Correction: Sibanda et al. Application of drone technologies in surface water resources monitoring and assessment: A systematic review of progress, challenges, and opportunities in the Global South. Drones 2021, 5, 84. Drones, 6(5), 131.
Lines 532. Explain the reason to survey every 6 years over 80 lakes. It might monitor it at least every two or three years with the methods provided in this manuscript.
Author Response

(The authors gave the same response as above.)
